# Coordination of biradial-to-radial symmetry and tissue polarity by HD-ZIP II proteins

Monica Carabelli[1,4], Luana Turchi[1], Giorgio Morelli [2], Lars Østergaard [3,5✉], Ida Ruberti[1,5,6] & Laila Moubayidin [3,4✉]

Symmetry establishment is a critical process in the development of multicellular organs and requires careful coordination of polarity axes while cells actively divide within tissues. Formation of the apical style in the Arabidopsis gynoecium involves a bilateral-to-radial symmetry transition, a stepwise process underpinned by the dynamic distribution of the plant morphogen auxin. Here we show that SPATULA (SPT) and the HECATE (HEC) bHLH proteins mediate the final step in the style radialisation process and synergistically control the expression of adaxial-identity genes, *HOMEOBOX ARABIDOPSIS THALIANA 3* (*HAT3*) and *ARABIDOPSIS THALIANA HOMEOBOX 4* (*ATHB4*). HAT3/ATHB4 module drives radialisation of the apical style by promoting basal-to-apical auxin flow and via a negative feedback mechanism that finetune auxin distribution through repression of *SPT* expression and cytokinin sensitivity. Thus, this work reveals the molecular basis of axes-coordination and hormonal cross-talk during the sequential steps of symmetry transition in the Arabidopsis style.

[1] Institute of Molecular Biology and Pathology, National Research Council, Rome, Italy. [2] Research Centre for Genomics and Bioinformatics, Council for Agricultural Research and Economics (CREA), Rome, Italy. [3] Department of Crop Genetics, John Innes Centre, Norwich, UK. [4] These authors contributed equally: Monica Carabelli, Laila Moubayidin. [5] These authors jointly supervised this work: Lars Østergaard, Ida Ruberti. [6] Deceased: Ida Ruberti. ✉email: lars.ostergaard@jic.ac.uk; laila.moubayidin@jic.ac.uk

Amajor challenge during organ morphogenesis includes the establishment of symmetries along the organ polarity axes (apical–basal, medio-lateral and dorso-ventral/ abaxial–adaxial)[1–5]. We have previously shown that the female reproductive organ (gynoecium) of *Arabidopsis thaliana* provides an excellent model for symmetry studies during organogenesis[6,7] (Fig. 1a). At early developmental stages, the ovary of the Arabidopsis gynoecium is bilaterally symmetric (one plane of symmetry) reflecting the congenital fusion of two carpels[8,9] (Fig. 1a). Later, a transition in symmetry occurs, when the apical end becomes radially symmetric[6] (three or more planes of symmetry) (Fig. 1), thus forming the style. We have shown that the overall bilateral-to-radial symmetry transition observed at the organ level is guided by consecutive transitions in the distribution of the plant hormone auxin, which includes a biradial state[6]. Initially, auxin accumulates in two lateral foci (two-foci stage) at the apex of the gynoecium in accordance with the inherent bilateral symmetry at this stage[6] (Fig. 1a). Subsequently, two new medial foci form (four-foci stage) establishing a biradially symmetric state of auxin distribution (two planes of symmetry at 90° to each other), before a continuous, ring-formed, auxin signalling maximum provides information for radial symmetry[6] (Fig. 1a). The requirement for this precise spatio-temporal pattern of auxin dynamics is well-established and controls the axisymmetric growth and patterning of tissues along the apical–basal and medio-lateral axes[5,6,10,11], during the developmental transitions

of the gynoecium. It is, however, still unresolved how this dynamic auxin pattern is coordinated with the adaxial–abaxial polarity axis during the radialisation process, within the radially symmetric style. In this paper, we present experiments suggesting a two-step mechanism whereby stage-specific transcription factor complexes control the timely expression of polarity identity genes to coordinate hormone distribution and sensitivity. The proposed mechanism therefore provides a framework for the integration of gene regulatory and hormonal activities during organogenesis.

## Results and discussion

**HAT3 and ATHB4 HD-ZIP II coordinate adaxial polarity during radial symmetry establishment in the Arabidopsis gynoecium.** *SPATULA* (*SPT*) encodes a basic helix–loop–helix (bHLH)-type transcription factor required for proper gynoecium development in *Arabidopsis*[12]. Compared to wild-type, gynoecia from *spt* mutants exhibit defects in symmetry transition and the apex remains unfused[6,12] (Fig. 1b). This split-style phenotype is due to the formation of a cleft in the medial region, and this defect is strongly exacerbated by combining the *spt* mutant with mutations in another bHLH transcription factor gene, *INDE-HISCENT* (*IND*)[13]. The SPT/IND module functions cooperatively to mediate their activities in organ development, at least partially by affecting auxin distribution[6,13,14]. In both *spt* and *ind spt* mutants, the stepwise auxin distribution is blocked at the lateral

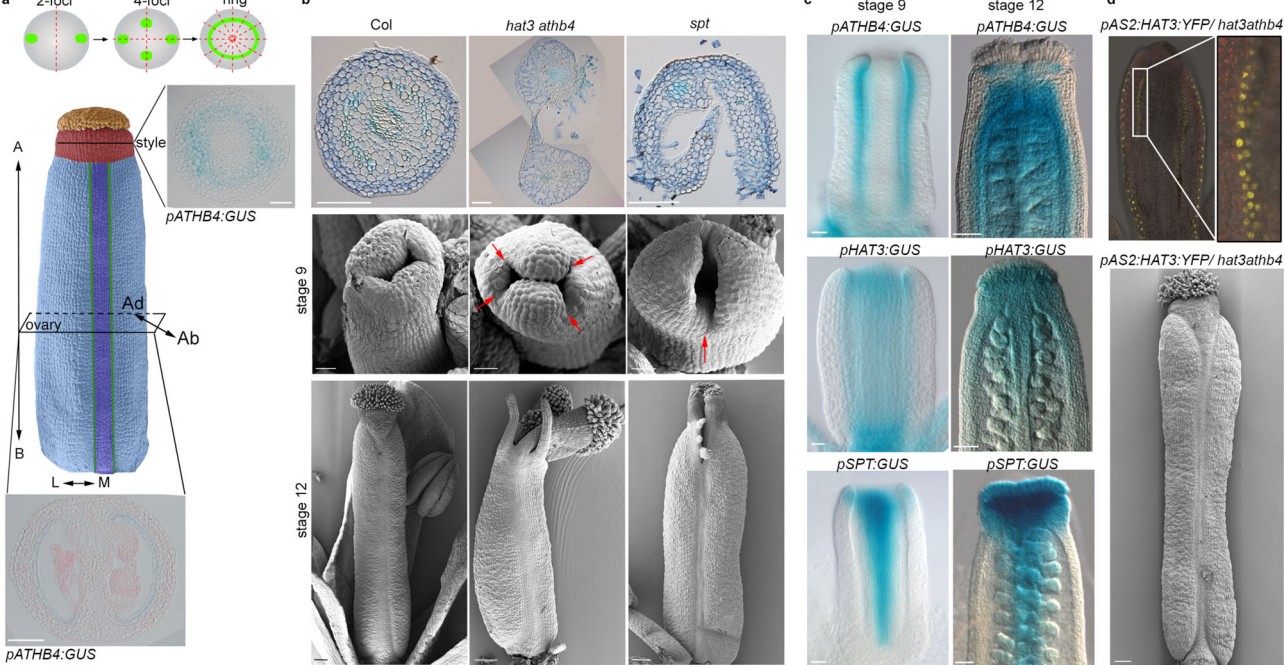

**Fig. 1 HD-ZIP II proteins HAT3 and ATHB4 coordinate adaxial/abaxial polarity axis during symmetry transition. a** Schematic representation of symmetry transition in Arabidopsis wild-type gynoecium (Col-0); (top) schematic indicating the dynamic distribution of the *DR5rev::GFP* signal in the outline of the gynoecium viewed from the top. Two-foci = bilateral symmetry—one plane of symmetry; four-foci = biradial symmetry—two planes of symmetry; ring = radial symmetry—≥3 planes of symmetry; (bottom) false-coloured scanning electronic micrograph (SEM) of wild type at stage 11. Stigma (orange), style (red), valves (light blue), valve margins (green), replum (violet), including GUS-stained cross-sections of the *pATHB4:GUS* reporter shown for radially symmetric style (top) and bilaterally symmetric ovary (bottom). Scale bars represent 50 µm. Double-headed arrows indicate the polarity axes: A apical, B basal, M medial, L lateral, Ab abaxial, Ad adaxial. **b** Optical images of Toluidine blue-stained cross-sections of stage-12 style (top panels; scale bars represent 100 µm) and SEM images (middle and bottom panels) of Col-0 (left), *hat3 athb4* (centre) and *spt* (right) at stage 9 (middle; scale bars represent 20 µm) and stage 12 (bottom; scale bars represent 100 µm). Red arrows indicate the different positions of the split style in the mutant backgrounds. **c** Optical images of GUS-stained gynoecia of *pATHB4:GUS* (top), *pHAT3:GUS* (middle) and *pSPT:GUS* (bottom) lines at stage 9 (left) and stage 12 (right) of gynoecium development. Scale bars represent 20 µm (left, stage 9) and 50 µm (right, stage 12). **b**, **c** Similar results were obtained from four independent experiments. Sections in (**a**, **b**) were made using samples from two of the four biological replicates in (**b**, **c**). **d** Confocal (top) and SEM (bottom) images of *pAS2:HAT3:YFP/hat3 athb4* gynoecia at stage 9 (top) and stage 12 (bottom) of development. Scale bar represents 100 µm. Similar results were obtained from three independent experiments.

two-foci stage, therefore failing to go through the four-foci stage and prevented from reaching the ring-formed auxin distribution required to complete the radialisation process[4].

SPT has previously been proposed to function in diverse biological processes by forming interactions with process-specific bHLH proteins. Among identified SPT interactors, bHLH transcription factors belonging to the HECATE (HEC) subfamily are required for gynoecium development[15,16]. Interestingly, scanning electron microscopy of a hec1,2,3 triple mutant showed that growth is retarded in the lateral region, leading to a style that is diagonally split (Supplementary Fig. 1). Moreover, the auxin distribution in this triple mutant reaches the four-foci stage (i.e. a step further in the distribution of auxin compared to spt and spt ind mutants), but it is blocked from completing the auxin ring[16]. In addition to SPT–HEC protein interactions, SPT and HEC genes also interact genetically[16]. However, the mechanistic significance of these interactions in the bilateral-to-radial symmetry transition of the gynoecium apex is unknown.

Two genes encoding HD-ZIP II family transcription factors were previously identified as direct downstream targets of both SPT and HEC proteins[17,18]. These two genes, HAT3 and ATHB4, are involved in leaf abaxial/adaxial patterning[19,20], shade-avoidance response[21], auxin-regulated developmental processes[22,23], and symmetry transition at the gynoecium apex as shown by the split-style phenotype of the hat3 athb4 double mutant[17]. Interestingly, we found this gynoecial split to occur diagonally with respect to the medio-lateral axis (Fig. 1b and Supplementary Fig. 2), suggesting that HAT3 and ATHB4 control symmetry transition in a spatial location different from SPT. The diagonal indentations in hat3 athb4 are similar to the split observed in gynoecia of the hec1/2/3 triple mutant (Supplementary Fig. 1), and a phenotype that also resembles loss-of-function mutants in the adaxial factors NUBBIN and JAGGED[24]. We therefore hypothesise that HAT3 and ATHB4 may have key roles downstream of the SPT/HEC module in coordinating the adaxial–abaxial axis during radial symmetry establishment, presumably via regulation of auxin dynamics. To sustain this role, we observed expression of both pHAT3:GUS ($n = 44/53$) and pATHB4:GUS[22] ($n = 34/40$) reporter lines in the gynoecium from early developmental stages, specifically restricted to the adaxial side of the two lateral valves (stage 9 in Fig. 1b and Supplementary Fig. 3a). Similar to SPT expression[25] ($n = 23/23$), the expression of HAT3 ($n = 45/47$) and ATHB4 ($n = 48/50$) was also detected in the medial tissues, including the apex of the gynoecium at later developmental stages (stage 12 in Fig. 1b and Supplementary Fig. 3a). Moreover, by analysing the pHAT3:HAT3:YFP and pATHB4:ATHB4:GUS transgenic lines in a wild-type background, we confirmed that the distribution of HAT3 and ATHB4 protein followed by the gene expression domains (Supplementary Fig. 3b). To corroborate the hypothesis that both factors function cell autonomously, at least partially, from the adaxial tissues, we performed tissue-specific complementation experiment whereby HAT3 was expressed under the control of the adaxial-specific ASYMMETRIC LEAVES 2 (AS2) promoter[26]. Our results showed that adaxially driven expression of HAT3 is sufficient to recover cotyledon, leaf, and gynoecium defects in the hat3 athb4 background (Fig. 1d and Supplementary Fig. 4). This suggests that the expression of HAT3 and ATHB4 in the adaxial domain is required for radialisation of the style and to maintain an adaxial/abaxial axis during the patterning process.

**SPT and HEC promote HAT3 and ATHB4 expression in the style and sustain organ radialisation**. We then tested whether HAT3 and ATHB4 expression is regulated by SPT and HEC1,2,3 transcription factors. Quantitative reverse transcription-polymerase chain reaction (qRT-PCR) experiments on RNA extracted from inflorescences revealed a significant reduction in both HAT3 and ATHB4 transcript levels in the hec1,2,3 triple mutant (Fig. 2a). This result was confirmed at the tissue-specific level using GUS reporter lines, which showed dramatic down-regulation of HAT3 and ATHB4 expression in hec single and multiple mutant gynoecia (Fig. 2b). While qRT-PCR experiments revealed a significant reduction in ATHB4 but not HAT3 expression in the spt mutant (Fig. 2a), both pHAT3:GUS ($n = 39/44$) and pATHB4:GUS ($n = 34/38$) reporters exhibited reduced expression in spt gynoecia (Fig. 2b). Moreover, pHAT3:GUS and pATHB4:GUS expression was undetectable in the spt hec1 double mutant ($n = 27/30$) and in spt hec1,2,3+/− ($n = 25/30$), respectively (Fig. 2b). These results are therefore in agreement with HAT3 and ATHB4 being synergistically regulated by SPT[17] and HEC1[18] during gynoecium development.

Since a genetic interaction between SPT and the HEC genes has been reported[16], we took a genetic approach to investigate the biological relevance of the SPT/HEC-mediated control of HAT3 and ATHB4 gene expression. We first crossed hat3 athb4 with spt and found that the triple hat3 athb4 spt mutant showed an additive effect, exhibiting both medial and diagonal clefts at their apices (Fig. 2c). If SPT and HECs mediate at least part of their functions in gynoecium formation via their synergistic regulation of HAT3 and ATHB4, it is possible that the ectopic expression of HAT3 and ATHB4 would rescue the gynoecium defects in spt mutant. We sought to test this by producing inducible overexpression lines of both HD-ZIP II genes. First, we found that inducing either of these two HD-ZIP II-encoding genes led to an increase in the frequency of radially symmetric styles in the spt mutant background (Fig. 2d, e and Supplementary Fig. 5). However, in contrast to the published effect of IND overexpression[14], ectopic induction of HAT3 and ATHB4 was insufficient to impose radial symmetry throughout the gynoecium (Supplementary Fig. 6a). Also, the rescue of the spt defects was not extended to the septum, which remained unfused similar to the spt control (Supplementary Fig. 5a). Interestingly, the simultaneous overexpression of HAT3 and ATHB4 in XVE::HAT3 × XVE::ATHB4 (F$_1$) seedlings enhanced the complexity in leaves by augmenting the leaf bilateral symmetry (Supplementary Fig. 6b), similarly to pin mutants[27], although not affecting gynoecium development (Supplementary Fig. 6a). Altogether, these data confirm that HAT3 and ATHB4 function downstream of the SPT/HEC module and reveal the relevance of this regulatory interaction at the gynoecium apex.

**HAT3 and ATHB4 facilitate radial symmetry by controlling polar auxin transport**. We next investigated whether HAT3 and ATHB4 coordinate radial symmetry establishment via regulation of auxin dynamics. During wild-type gynoecium development, auxin accumulation is highly dynamic as shown by the analysis of the auxin signalling reporter DR5rev::GFP (Fig. 3a)[6,10,11]. The spt mutant fails to accumulate auxin in the medial foci, leading to a 'split-style' phenotype in the medial apical part of the young gynoecium[6] (Figs. 1b and 3a). At early stages of hat3 athb4 development, the establishment of the two lateral foci was not affected (Fig. 3a). These foci are important to sustain apical–basal growth, and accordingly, this mutant exhibited correct bilateral symmetry in the ovary (Supplementary Fig. 7). However, while the spt mutant is blocked at the two-foci stage, the hat3 athb4 mutant can advance to the four-foci stage, but fails to establish the ring-formed auxin maximum at the gynoecium apex (Fig. 3a). Interestingly, the medial DR5rev::GFP foci are already observed at stages 5 and 6 in the hat3 athb4 mutant gynoecia ($n = 17/19$ samples; Fig. 3a), whereas they appear at stages 8 and 9 in wild type ($n = 20/24$; Fig. 3a).

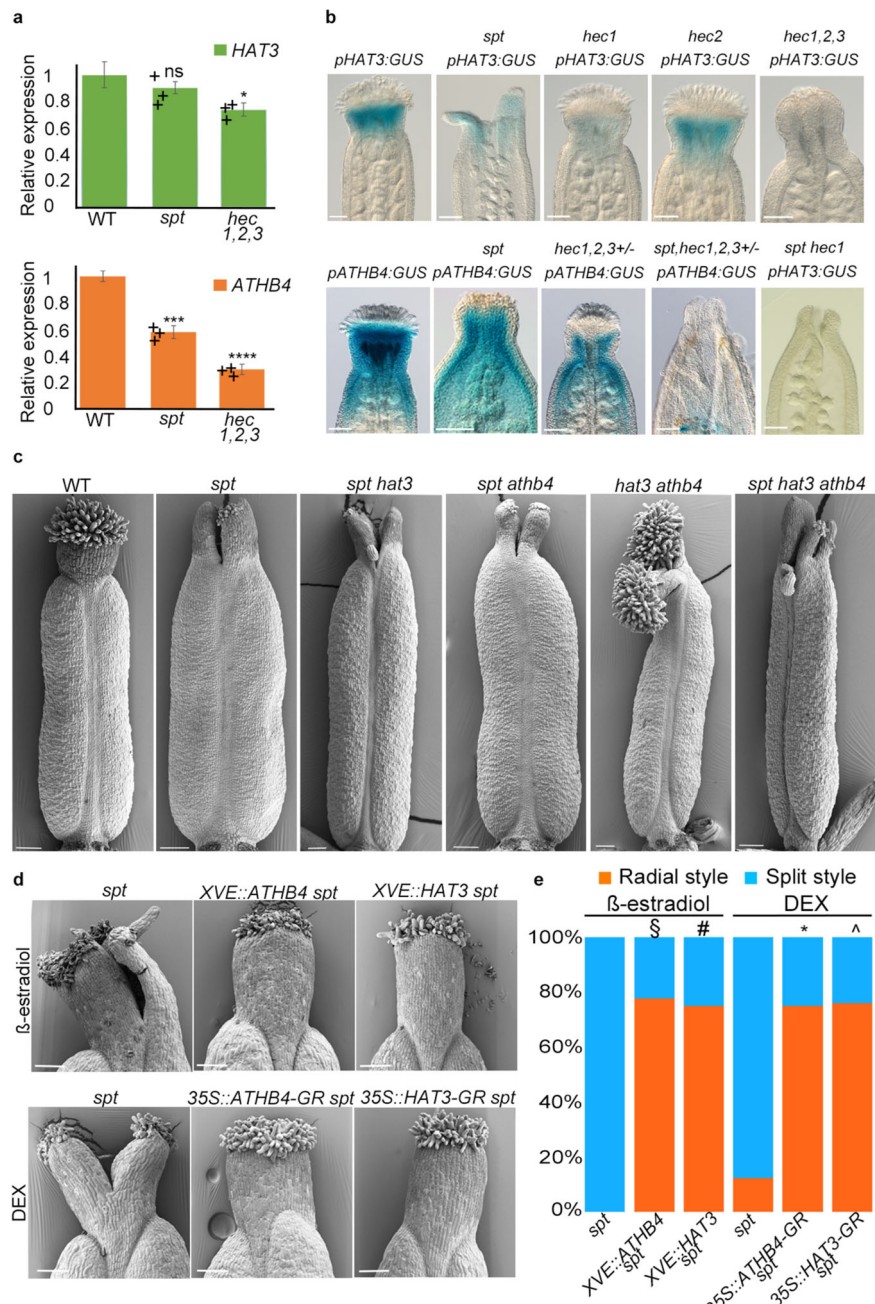

**Fig. 2 SPT/HEC synergistic control of *HAT3* and *ATHB4* expression and genetic analyses. a** qRT-PCR of *HAT3* and *ATHB4* in Col-0, *spt* and *hec1,2,3* mutant inflorescences normalised against *UBIQUITIN10*. The centre for the error bars represents mean, while error bars represent SD; *$p < 0.05$, ***$p < 0.001$, ****$p < 0.0001$, ns indicates no statistically significant difference (two-sided Student's *t* test), $n = 3$ technical replicates. The panel displays similar results obtained from one of three biological replicates analysed, and up to six technical replicates for each gene and for each biological replicate were performed. **b** Optical images of stage-12 gynoecia showing *pHAT3:GUS* (top) and *pATHB4:GUS* (bottom) expression in Col-0, *spt* and combinations of *hec1,2,3* and *spt* mutants, as depicted on individual panels. **c** SEM images of stage-12 gynoecia of Col-0, *spt*, *spt hat3*, *spt athb4*, *hat3 athb4* and *spt hat3 athb4*. Scale bars represent 100 μm. **b**, **c** Similar results were obtained from three independent experiments. **d** SEM images of stage-13 styles of *ATHB4* (*XVE::ATHB4* and *35S::ATHB4:GR*) and *HAT3* (*XVE::HAT3* and *35S::HAT3:GR*) overexpression lines in *spt* background, treated with either 20 μM β-estradiol (top) or 10 μM DEX (bottom). Scale bars represent 100 μm. **e** Quantification of the radial (orange bars) and split (blue bars) style phenotype of *HAT3* and *ATHB4* overexpression lines (*XVE::HAT3*, *XVE::ATHB4*, *35S::HAT3:GR* and *35S::ATHB4:GR*) in *spt* background treated with mock, β-estradiol and DEX. Phenotypic classes were compared using contingency 2 × 2 tables followed by Pearson's $\chi^2$ test. Two-tailed *P* values are as follows: *spt-12 XVE::ATHB4* β-estradiol vs *spt-12* β-estradiol *P* < 0.00001 (§); *spt-12 XVE::HAT3* β-estradiol vs *spt-12* β-estradiol *P* < 0.00001 (#); *spt-12 35S::ATHB4:GR* DEX vs *spt-12* DEX *P* < 0.00001 (*); *spt-12 35S::HAT3:GR* DEX vs *spt-12* DEX *P* < 0.00001 (^). *P* values < 0.01 were considered as extremely statistically significant.

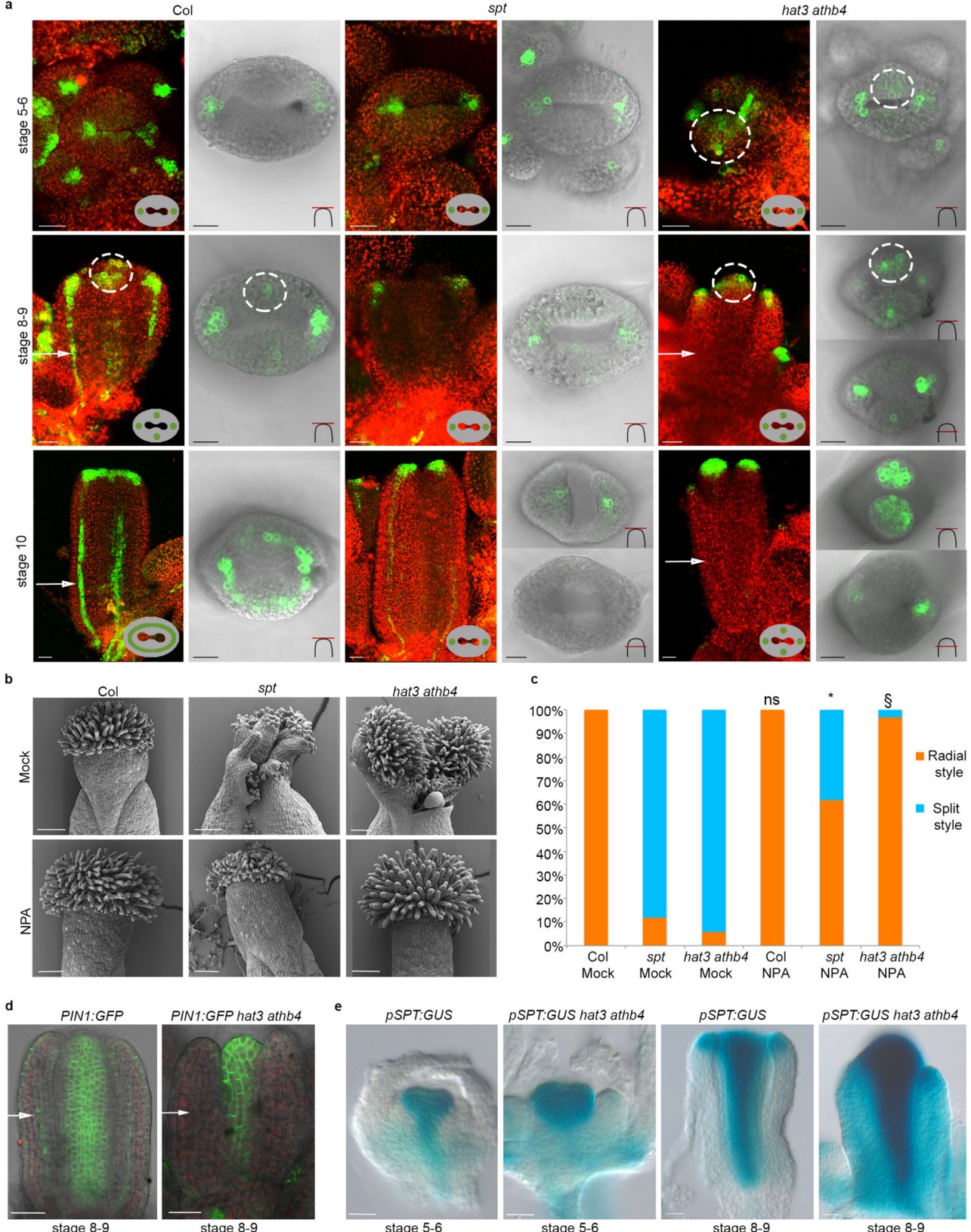

We therefore next tested whether the defect in auxin distribution is causative of the *hat3 athb4* gynoecium phenotype. To this end, wild-type *hat3 athb4* and *spt* mutants were treated with 1-naphthylphthalamic acid (NPA), an inhibitor of polar auxin transport[28]. NPA has previously been shown to inhibit auxin transport from the gynoecium base towards the apex, thereby causing developmental defects along the apical–basal axis of the gynoecium, leading to a reduction in the size of the ovary[5] (Supplementary Fig. 7). In contrast, the application of NPA was previously reported to rescue the split style of the *spt* mutant[5] as well as that of other mutants with apical defects[29]. Under our experimental conditions, 100 μM NPA spray applications were able

**Fig. 3 HAT3 and ATHB4 control auxin dynamics allowing a biradial-to-radial symmetry transition of auxin distribution. a** Confocal images of *DR5rev:: GFP* in Col-0 (left), *spt* (centre) and *hat3 athb4* (right) at stages 5 and 6 (top), stages 8 and 9 (middle) and stage 10 (bottom) of gynoecium development. In the side view pictures (black background), chlorophyll autofluorescence is shown in red, while the gynoecium outline indicates the top-view of auxin dynamics. In the merged confocal/bright field top view pictures (light background), the schematic drawings in the bottom right corner indicate the confocal plane of the images. Dashed circles indicate the medial auxin foci. White arrows indicate the adaxial valve tissues. Scale bars represent 20 μm. Similar results were obtained from four independent experiments. **b** SEM images of Col-0 (left), *spt* (centre) and *hat3 athb4* (right) at stage 12 treated with mock (top) or 100 μM NPA (bottom). Scale bars represent 100 μm. **c** Quantification of the radial (orange bars) and split (blue bars) style phenotype of Col, *spt* and *hat3 athb4* treated with 100 μM NPA or mock. Phenotypic classes were compared using 2 × 2 contingency tables followed by Pearson's $\chi^2$ test. Two-tailed *P* values are as follows: Col NPA vs Col mock, *P* = 1 (ns, no statistically significant difference); *spt-12* NPA vs *spt-12* mock, *P* < 0.00001 (\*); *hat3 athb4* NPA vs *hat3 athb4* mock, *P* < 0.00001 (§). *P* values < 0.01 were considered as extremely statistically significant. Similar results were obtained from three independent experiments. **d** *PIN1:GFP* expression in wild-type (left) and *hat3 athb4* (right) gynoecia at stages 8 and 9 of development. White arrows indicate the adaxial valve tissues. Scale bars represent 50 μm. **e** *pSPT:GUS* expression in wild-type and *hat3 athb4* gynoecia at stages 5 and 6 (left) and stages 8 and 9 (right) of development. Scale bars represent 20 μm. **d**, **e** Similar results were obtained from two independent experiments.

to complement 26 of the 42 *spt* mutant styles analysed (61.9%) (Fig. 3b, c). Moreover, the NPA applications rescued the split-style defects in 32 out of 33 *hat3 athb4* double mutants (96.9%) (Fig. 3b, c). In addition to the style, *hat3 athb4* showed hypersensitivity to NPA application also along the apical–basal axis, as shown by the strongly decreased ovary when compared to wild type (Supplementary Fig. 7), suggesting that the main downstream activity of HAT3 and ATHB4 is to control auxin distribution via regulation of the polar auxin transport. This is in line with the previously identified role of *HAT3* and *ATHB4* as coordinators of apical embryo development via regulating the PIN1-mediated auxin accumulation[22,23]. Therefore, we next analysed *PIN1:GFP* expression in wild-type and *hat3 athb4* gynoecia and confirmed a strong reduction in the number of mutant organs showing fluorescent signal specifically in the adaxial tissues of the valves. Only 9 out of 46 *hat3 athb4* gynoecia analysed (19.5%) retained *PIN1:GFP* expression in the valves compared to 29 of the 33 wild-type gynoecia (88%) (Fig. 3d). These data correlate with a strong reduction of *DR5rev:: GFP* in the *hat3 ahtb4* valves (Fig. 3a), and hypersensitivity of the double mutant to NPA application (Supplementary Fig. 7), confirming a role for HAT3 and ATHB4 in controlling auxin distribution in the gynoecium.

**HAT3 and ATHB4 repress *SPT* expression and coordinate growth along polarity axes.** The presence of medial auxin foci in the *hat3 athb4* double mutant correlates with proper development of the medial style, whereas lack of medial foci in the *spt* mutant leads to a split in this position (Fig. 1b). Interestingly, the premature emergence of the medial foci in *hat3 athb4* double mutant (Fig. 3a) led us to hypothesise a negative feedback loop actioned by the adaxial regulators HAT3 and ATHB4 on the medial determinant, SPT, enabling growth at the medial axis of the HD-ZIPs-II double mutant apexes (Supplementary Fig. 2). Thus, we analysed the expression of the *pSPT:GUS* reporter in the *hat3 athb4* mutant compared to wild type and observed a stronger expression of *SPT* at early developmental stages of the *hat3 athb4* mutant gynoecia (n = 25/27) (Fig. 3e). Comparing gynoecia from the *hat3 athb4* double mutant to the *hat3 athb4 spt* triple mutant confirms that *SPT* upregulation in the *hat3 athb4* background contributes to the sustained radial growth in the medial–apical region of the mutant gynoecia (Fig. 2c). Altogether, whilst demonstrating that HAT3 and ATHB4 control auxin distribution in the gynoecium apex, these data show that HAT3 and ATHB4 HD-ZIP II proteins are required to coordinate growth along the medio-lateral and adaxial–abaxial polarity axes during radialisation of the style, functioning at least partially in a cell-autonomous manner from adaxial domains (Fig. 1d). Our results show that *HAT3* and *ATHB4* are key coordinators of adaxial polarity during organ symmetry establishment. Their role is sustained via precise spatio-temporal expression patterns,

promoting dynamic auxin distribution and ultimately fine-tuning organ shape and function.

**HAT3 and ATHB4 control cytokinin sensitivity downstream of SPT/HEC.** SPT and HECs are key regulators of the delicate balance between auxin and cytokinin required for gynoecium patterning[16,30–32]. We therefore tested whether HAT3 and ATHB4 contribute to this function. In agreement with previous results, 100% of the *hec1,2* (n = 80) and *spt* (n = 80) gynoecia showed hypersensitivity to cytokinin application[16], leading to apically unfused organs and extensive tissue proliferation at their tip (Fig. 4a, b). In *hat3 athb4* mutant gynoecia, we observed an intermediate effect with outgrowths of split apices in the diagonal part (48 of 80 gynoecia analysed), while no changes were observed at the medial region (Fig. 4a, b), where *SPT* is upregulated (Fig. 3e). These data suggest a role for HAT3 and ATHB4 in modulating cytokinin levels downstream of SPT/HEC in a spatially specific manner. Moreover, cytokinin treatments led occasionally to trichome formation on the abaxial side of *hat3 athb4* valves, which was not observed in cytokinin-treated wild-type gynoecia (Fig. 4c and Supplementary Fig. 8). This phenotype has been associated with cytokinin hypersensitivity[33–35], and suppression of trichome fate at the abaxial epidermis of the gynoecium was shown to be controlled by the abaxial-identity factor, KANADI1, in a cytokinin-regulated fashion[33]. Taken together, these results indicate that HAT3 and ATHB4 contribute to controlling the hormonal balance between auxin and cytokinin to coordinate the body-axis formation and symmetry establishment downstream of SPT and HEC proteins (Fig. 4d).

In conclusion, our work reveals a mechanism for coordinating symmetry transition and axis polarity to complete the bilateral-to-radial transition of the Arabidopsis style. We propose a model in which SPT functions with process-specific bHLH transcription factors in two consecutive steps to control auxin distribution (Fig. 4d). First, SPT and IND facilitate a bilateral-to-biradial transition by promoting the formation of auxin maxima in a four-foci biradial pattern[7]. Subsequently, SPT in concert with HEC proteins control the biradial-to-radial symmetry transition, thus establishing the ring-shaped auxin maximum and restricting cytokinin sensitivity (Fig. 4d). This regulatory module also controls the expression of the adaxial regulators *HAT3* and *ATHB4* to coordinate the adaxial–abaxial and medio-lateral polarity axes, thereby providing a molecular framework to facilitate bilateral-to-radial symmetry transition of the style required for successful reproduction.

## Methods

**Plant materials and growth conditions.** Plants were grown on soil in long-day conditions (16 h light/8 h dark) in controlled environment rooms. The following loss-of-function mutant lines *spt-12*[36] (WISCDSLOX386E06), *hat3 athb4*[22] (Salk_014055; Salk_104843), and *hec1,2,3*[16] (GK_297B10; SM_3_17339; SALK_005294) and overexpressing *XVE::HAT3*[22] line were in Col-0 background,

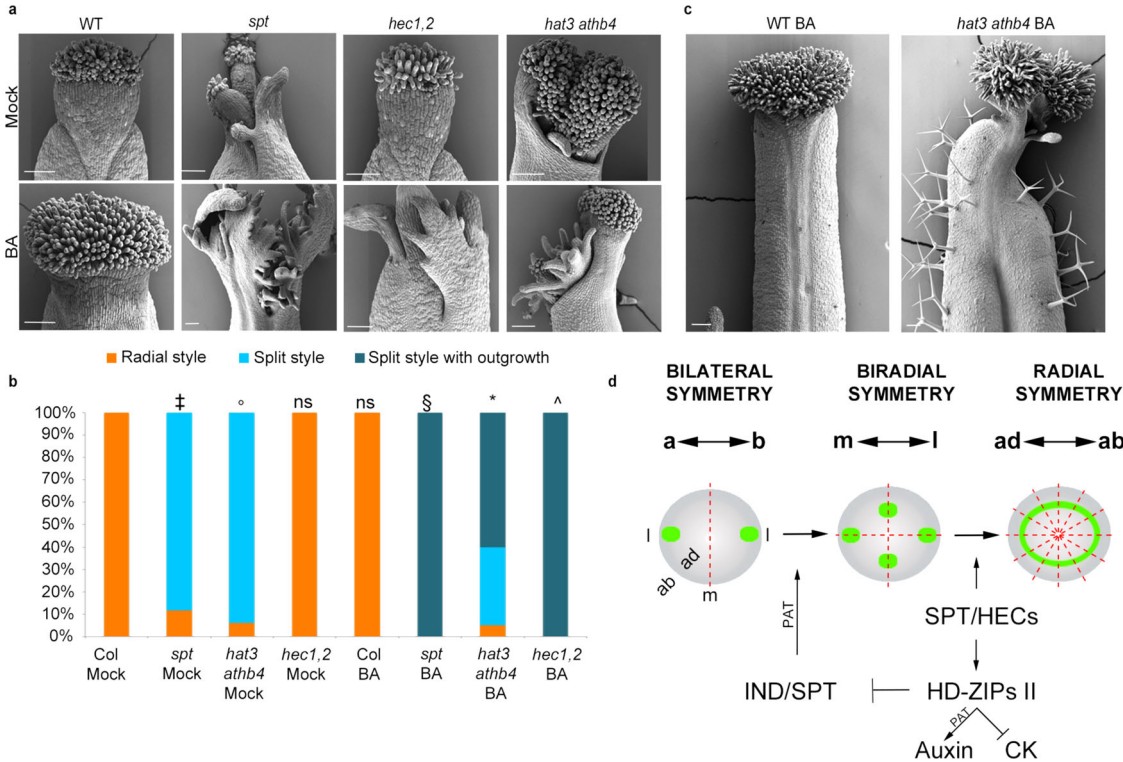

**Fig. 4 HAT3 and ATHB4 suppress sensitivity to cytokinin. a** SEM images of stage-13 gynoecia of Col-0 wild-type, *spt* single mutant, *hec1,2* and *hat3 athb4* double mutants treated with mock (top) or 50 µM of cytokinin (BA) (bottom). Scale bars represent 100 µm. **b** Quantification of cytokinin (BA) effect on gynoecium apex development of genotypes depicted on the graph. Phenotypic classes were compared using 3 × 2 contingency tables followed by Pearson's $\chi^2$ test. Two-tailed *P* values are as follows: *spt-12* Mock vs Col Mock, *P* < 0.00001 (‡); *hat3 athb4* Mock vs Col Mock *P* < 0.00001 (°); *hec1,2* Mock vs Col Mock *P* = 1 (ns, no statistically significant difference); Col BA vs Col mock, *P* = 1 (ns); *spt-12* BA vs *spt-12* mock, *P* < 0.00001 (§); *hat3 athb4* BA vs *hat3 athb4* mock, *P* < 0.00001 (*); *hec1,2* BA vs *hec1,2* mock, *P* < 0.00001 (^). *P* values < 0.01 were considered as extremely statistically significant. **c** SEM images of stage-13 gynoecia of wild-type (left) and *hat3 athb4* (right) treated with 50 µM of BA, showing the occasional trichome formation on the valves of the hypersensitive double mutant. Scale bars represent 100 µm. Similar results were obtained from two independent experiments. **d** Model showing the genetic regulation of consecutive transitions of auxin signalling (green) at the gynoecium apex (grey outline—top view). Double-head arrows represent the polarity axis; a apical domain, b basal domain, m medial domain, l lateral domain, ab abaxial domain, ad adaxial domain. Dashed red lines represent the planes of symmetry along the medio-lateral axis. PAT polar auxin transport, CK cytokinin.

and reporter lines of *pATHB4:GUS*[22], *pSPT:GUS*[25], *PIN1:GFP*[37], and *DR5rev:: GFP*[37] were employed.

**DNA constructs.** *XVE::ATHB4* was constructed as follows: the complete CDS of *ATHB4* (TAIR, AT2G44910) was amplified with *Apa*I and *Spe*I ends using the primers AB4-S: AATTGGGCCCATGGGGGAAAGAGATGAT and AB4-AS: GGATACTAGTCTAGCGACCTGATTTTTGCT and cloned into the XVE vector *pER8*[38] in order to obtain an ER-inducible form of ATHB4. To generate the *pHAT3:GUS* transcriptional fusion, a DNA fragment spanning from −2094 to +3 (ATG) of *HAT3* (At3G60390) was amplified using the primers HT3-BP1, GGGG ACAAGTTTGTACAAAAAAGCAGGCTTCGAGCGTTGAATAAGTGTGTTAA and HT3-BP2, GGGGACCACTTTGTACAAGAAAGCTGGGTCCATTTTTCTC AACCCCAGAA, and cloned with Gateway™ technology into the *pMDC163* vector[38]. The *pHAT3:HAT3:YFP* translational fusion was constructed as follows: first, the putative promoter of *HAT3* was extended from −1768 to −5066 bp in the construct *pHAT3:HAT3:GFP pMDC107*[22]. Then, an intermediate construct was generated in *pBKS II* plasmid by separate rounds of PCR and cloning, containing the *HAT3* gene from *Bst*BI unique site to the stop codon (excluded), a 9x alanine linker, the *EYFP* CDS, and a 3′*UTR* portion of *HAT3*. Finally, the fragment *HAT3: ala9:EYFP-3′* HAT3 was cut with *Bst*BI and *Kpn*I and introduced into the plasmid generated in the first step. Amplifications were performed using the primers: HAT3 *Pme*I for, AAGTTTAAACGATGTATTGTCCTGATGGTTC and HAT3 *Pac*I rev, GGTTAATTAATGATGACTCCATATATG; *Spe*I HAT3 3′utr fwd, AAAC*- TAGTCTAAAGATAATAGGTTTGGTGATTTG (* indicates a mutation intro-duced in the original HAT3 sequence to generate the *Spe*I site) and *Kpn*I HAT3 3′ utr rev, AAGGTACCTCCAGTTCGATTGGTAGAAAC.

The *pATHB4:ATHB4:GUS* translational fusion was constructed as follows: first, a DNA fragment containing the *ATHB4* genomic sequence spanning from −1990 to +1635 bp was amplified from genomic DNA (Col-0) and fused to GFP in the *pMDC107* vector using Gateway™ technology. Primers on the genomic DNA were as follows: −1990 ATHB4 F, CTACACTCTCTCCCACACCATTCACA and

+1635 ATHB4 R, GCGACCTGATTTTTGCT. Then, a genomic fragment from −4874 to the *Pac*I site at −1148 of the genomic *ATHB4* sequence was amplified from genomic DNA and added to the previous construct, exploiting the *Pme*I site in *pMDC107* and *Pac*I site in the *ATHB4* gene, thus extending the putative *ATHB4* promoter up to −4874. Then, an intermediate construct was generated in *pBluescript II* plasmid by separate rounds of PCR and cloning: first, the *ATHB4* gene from +1422 bp, including the *Bbv*CI site to the stop codon (excluded), was amplified from genomic DNA inserting a 9x alanine linker. Primers used were as follows: +1422 ATHB4 F, AAGAGCTCGCTGAGGGCGTTGAAGTTG and +1635 ATHB4 R, AACGCGGGCCGCAGCAGCTGCCGCAGCTGCGCGACCTGA TTTTTGCTGG. Second, the *GUS* CDS was amplified with *Not*I and *Bam*HI extremities from *pBI121* plasmid and cloned in the *pBluescript II* intermediate downstream of the *ATHB4* fragment. Third, a 3′*UTR* portion of *ATHB4* was amplified from genomic DNA and cloned downstream of the *GUS* CDS in the intermediate plasmid. Primers were as follows: *Xcm*I-GUS_TAG-3′UTR, CCCGG GCCACCGGCGGCATGGAGGGAGGCAAACAATAGGGAAGGAGTATCTTC GGT and 3′UTR-X2-*Eco*RI, CCCGGGGATATCCCACCATGTTGGGTTGAATT GAACATTAAGAGACTG. Finally, the fragment *ATHB4:ala9:GUS-3′UTR ATHB4* was cut with *Bbv*CI and *Bst*XI from the *pBluescript II* intermediate and introduced into the vector generated in the first step, cut with the same enzymes, thus substituting the *GFP* marker with the *GUS* marker and the 3′*UTR* of *ATHB4*. The *pATHB4:ATHB4:GUS* construct was stably transferred in Col-0.

The *35S::HAT3:GR* and *35S::ATHB4:GR* were constructed as follows: the *GUS* gene was removed from the *pBI121* vector using the *Bam*HI–*Sst*I restriction sites, replaced with the *GR* sequence and then amplified with *Bam*HI–*Sst*I restriction ends from the *pTA7002* vector[39]. This vector was then cut with *Xba*I–*Bam*HI and used to insert either *HAT3* cDNA or *ATHB4* cDNA amplified with *Xba*I–*Bam*HI extremities. Primers used for *HAT3* cDNA were as follows: HAT3 *Xba*I fwd, CG GCTCTAGAATGAGTGAAAGAGATGATGG and HAT3 *Bam*HI rev, AAATG GATCCATATGAGAACCAGCAGG. Primers used for *ATHB4* cDNA were as follows: ATHB4 *Xba*I fwd, CGGCTCTAGAATGGGGGAAAGAGATGATGG and ATHB4 *Bam*HI rev, AAATGGATCCATGCGACCTGATTTTTGCTG. The

*pAS2:HAT3-YFP* was constructed as follows: a genomic region containing the putative *AS2* promoter from −3990 to −1 was amplified with Gateway™ technology and cloned into *pDONR-Zeo* vector. Before the AttB2 sequence, *Bam*HI and *Sac*II sites were added. Separately, an intermediate construct was generated in *pBKS II* plasmid by separate rounds of PCR and cloning, containing the *HAT3* cDNA, a 9x alanine linker and the *EYFP* CDS. The *HAT3:YFP* fusion was then inserted downstream of the *AS2* promoter in the pDONR-Zeo plasmid using the *Bam*HI-*Sac*II sites. The *pAS2:HAT3:YFP* fragment was then introduced into the *pMDC99* vector[40]. Primers for *AS2* promoter were as follows: pAS2 fwd, TGGT AGCTAGCGTTGTTGACA and pAS2 rev, TTTAATGACTTGAAAATGGAG. All transgenic lines used were in the Col-0 background, with the exception of *pAS2: HAT3:YFP*, which was stably transformed in the *hat3 athb4* background. The *pHAT3:HAT3:YFP* and the *pATHB4:ATHB4:GUS* constructs were able to completely rescue the *hat3 athb4* mutant phenotype (70 *hat3 athb4 HAT3::HAT3: YFP* lines out of 70 and 50 *hat3 athb4 pATHB4::ATHB4:GUS* lines out of 50 showed wild type-like gynoecia).

**Primers for genotyping.** HAT3 WT:

HAT3 unC: GAAACTGTACTGCTGCACAAGTG

HAT3 revC: CTTCTACTCTACCTCCTCCAACTGC

*hat3-3* mutant:

LBb1 GCGTGGACCGCTTGCTGCAACTC

in combination with HAT3 unC

ATHB4 WT:

ATHB4 unA: GATTGGGCAGAAGCAAAAGCTGAGGGCAAG

ATHB4 revA: CCCTAGCGACCTGATTTTTGCTGG

*athb4-1* mutant:

LBb1 GCGTGGACCGCTTGCTGCAACTC

in combination with ATHB4 revA

HEC1 WT:

HEC1 WT fwd: TAGAAGGGAGAGAATAAGCGAG

HEC1 WT rev: AATGAACACAAGCCTGATAGC

*hec1* mutant (GABI-KAT line 297B10):

*hec1* MUTANT fwd: ACCACAACAACACTTACCCTTTTC

*hec1* MUTANT rev: ATATTGACCATCATACTCATTGC

HEC2 WT:

HEC2 WT fwd: CTCACAAAACCTTAACTAGATGTCTGA

HEC2 WT rev: ATGCTTTCTGAATCCAACACCC

*hec2* mutant (T-DNA SM_3_17339):

*hec2*- MUTANT fwd: CCGACACTCTTTAATTAACTGACACTC

in combination with HEC2 WT rev primer

HEC3 WT:

HEC3 WT fwd: TCTTTATTTTTTCTCCGAACCA

HEC3 WT rev: AAGCCGTATCCATTTTAGTGCC

*hec3* mutant (SALK_005294 line):

*hec3* MUTANT fwd: GTTCACGTAGTGGGCCATC

in combination with HEC3 WT rev primer

SPT WT

SPT WT fwd: GAAGAAGCAGAGAGTGATGGGAGA

SPT WT rev: TGACTTGGAAGAGGGAGCTTCA

*spt-12* mutant:

p745: AACGTTCCGCAATGTGTTATTAAGTTGTC

in combination with SPT WT fwd primer

**Toluidine blue staining.** Tissues were fixed for 16 h at 25°C in 3.7% formaldehyde, 5% acetic acid, and 50% ethanol and subsequently dehydrated through an ethanol series until 70%. The tissues were embedded in paraffin. An RM 2125 rotary microtome (Leica) was used to make 10 μm transverse sections of Col, *spt-12* and *hat3 athb4* gynoecia at stage 12. Paraffin was removed from the sections by two rounds of incubation in 100% Histoclear (National Diagnostics) for 10 min at room temperature, followed by two washes in 100% ethanol for 2 min at room temperature, air-dried for 30 min and stained for 10 min by an aqueous solution containing 0.005% Toluidine blue O (Acros Organics). Slides were washed for 1 min in water; sections were mounted in a histological mounting medium Histo-mount (National Diagnostic) and examined under Leica DM600 light microscopy. Images were taken using the Leica LAS AF7000 software.

**Gynoecium and seedling treatments.** Col-0, *spt-12* and *hat3-3 athb4-1* gynoecia were treated with 100 μM NPA (Duchefa Biochemie N0926), 50 or 100 μM BA (6-benzylaminoadenine, Sigma B3408) or mock, as previously described[5,16]. Seedlings resulting from an F1 *XVE::HAT3* × *XVE::ATHB4* cross were grown for 15 days on MS medium supplemented with 20 μM β-estradiol (Sigma, E8875) or mock. Inflorescences resulting from an F1 *XVE::HAT3* × *XVE::ATHB4* cross, as well as *XVE::HAT3/WT*, *XVE::HAT3/spt*, *XVE::ATHB4/WT* and *XVE::ATHB4/spt* inflorescences were sprayed with 20 μM β-estradiol three times every second day and gynoecia were fixed after 5 days from the last treatment. *35S::HAT3-GR/spt* and *35S::ATHB4-GR/spt* inflorescences were sprayed with 10 μM dexamethasone (DEX) three times every second day and gynoecia were fixed after 5 days from the last treatment. All spray treatments used 0.01% final concentration of Silwet L-77.

**Scanning electron microscopy.** Gynoecia and leaves were fixed for 16 h at 25 °C in 3.7% formaldehyde, 5% glacial acetic acid and 50% ethanol. After complete dehydration through an ethanol series until 100%, gynoecium and leaves were critical point dried. Samples were dissected and coated with gold and examined under Zeiss Supra 55VP field emission scanning electron microscope using an acceleration voltage of 3 kV. The SmartSEM software (Zeiss) was used to operate the microscope and collect the images. Gynoecia from distinct inflorescences and leaves of seedlings of several biological independent genotypes were observed and counted for their phenotype.

**RNA extraction and qRT-PCR.** Three independent biological extractions of total RNA were isolated from Col-0, *hat3 athb4* and *spt-12* inflorescences using RNeasy Plant Mini Kit (Qiagen), including treatment with RNase-free DNase (Qiagen) following the manufacturer's instructions. Each RNA sample was reverse transcribed using the M-MLV Reverse Transcriptase (Promega) according to the manufacturer's instructions. qRT-PCR experiments were performed in triplicates from each RNA sample using BRYT Green-based GoTaq qPCR Master Mix (Promega) with Chromo4 Real-Time PCR Detection System (Bio-Rad). Expression levels were calculated relative to *UBIQUITIN 10* using the $2^{-\Delta\Delta ct}$ method. Primers were designed according to the recommendations of Applied Biosystems. Fifteen technical replicates for each gene were carried out using three biological replicates for each genotype. Representative results from one biological replicate are shown in the figure. Analysis was conducted using the gene-specific primers listed below:

for *HAT3*: HAT3 fwd, GCATCTTCATCACACATGCAG and HAT3 rev, CGATCTCATGTCCGAGTTTCT

for *ATHB4*: ATHB4 fwd, GCTTGAATCTTATGCCGTTGA and ATHB4 rev, TGTGCATGTGTTGAAACGAA

for *UBIQUITIN 10*: UB10 fwd, AGAACTCTTGCTGACTACAATATCCAG and UB10 rev, GTTAAGACGTTGACTGGGAAAACTAT

**GUS histochemical assay.** To visualise *pHAT3:GUS*, *pATHB4:GUS*, *pATHB4: ATHB4:GUS* and *pSPT:GUS* lines, GUS histochemical assay was performed using 1 mg/ml of β-glucuronidase substrate X-Gluc (5-bromo-4-chloro-3-indolyl glucuronide, Melford) dissolved in dimethyl sulfoxide. X-Gluc solution contains 100 mM sodium phosphate buffer, 10 mM EDTA, 0.5 mM $K_3$ Fe(CN)$_6$, 3 mM $K_4$Fe(CN)$_6$, 0.1% Triton X-100 according to the JIC standard operating procedures. Wild-type and mutant inflorescences of *pHAT3:GUS*, *pATHB4:GUS* and *pSPT:GUS* were pretreated for 1 h with acetone at −20 °C, washed two times for 5 min in 100 mM sodium phosphate buffer, washed for 30 min in 100 mM sodium phosphate buffer containing 1 mM $K_3$–$K_4$ at room temperature and then incubated between 4 and 6 h at 37 °C in the X-Gluc solution. To visualise *pSPT:GUS* in Col and *hat3 athb4* background, samples were stained for 5 h. After staining, the samples were washed with water and then replaced with 70% ethanol until chlorophyll was completely washed out from the samples. Gynoecium was dissected, mounted in chlorohydrate (Sigma) solution and analysed using either a Leica DM6000 light microscopy (Leica LAS AF7000 software) or a Zeiss Axioscope2 Plus (ZEN 2012 software).

**Confocal microscopy.** Confocal microscopy analyses were performed on an Inverted Z.1 microscope (Zeiss, Germany) equipped with Zeiss LSM 700 spectral confocal laser scanning unit (Zeiss, Germany), using a ZEN 2012 software. GFP and EYFP samples were both excited by a 488 nm, 10 mW solid laser with emission at 492–539 nm[22]. For the lateral view of gynoecia, floral buds were dissected and mounted in water. For the top views of style region, gynoecia were dissected, mounted vertically in an agar dish and observed also using transmitting light (bright field)[6].

**Statistical analysis.** Phenotypic classes were compared using contingency tables (either 2 × 2 or 2 × 3) followed by Pearson's $\chi^2$ test. Two-tailed *P* values < 0.0001 were considered extremely statistically significant. Statistical analysis was performed using experimental numbers obtained by counts, while graphs in figures show the percentage of the results.

Figures' panels were assembled using Photoshop®.

**Reporting summary.** Further information on research design is available in the Nature Research Reporting Summary linked to this article.

# Data availability
All processed data are contained in the manuscript or in the Supplementary information. Biological material and data from this study will be available upon request and with no restrictions. Source data are provided with this paper.

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

## Acknowledgements

With this work, we would like to remember Ida Ruberti who sadly passed away during the late stage of this project, for her instrumental role in the conceptualisation of the project and the many great discussions of the results. The work was supported by the Royal Society University Research Fellowship URF\R1\180091 (to L.M.), by the UKRI Biotechnological and Biological Sciences Research Council Grant BB/M004112/1 (to L.Ø.), by the Institute Strategic Programme grant (BB/P013511/1) to the John Innes Centre from the UKRI Biotechnological and Biological Sciences Research Council, by the Italian Ministry of Education, University and Research, PRIN Programme 2010HEBBB8_004 (to I.R.), and by the Italian Ministry of Agricultural, Food and Forestry Policies, BIOTECH Programme D.M. 15924 (to G.M.). We thank Prof. Jan Lohmann (University of Heidelberg, Germany) for providing the *hec1,2,3* mutant and Yrjö Helariutta for discussions and helpful comments on the manuscript.

## Author contributions

L.M., L.Ø. and I.R. conceptualised the project, L.M. designed the experimental research with input from G.M., L.Ø. and I.R. L.M., M.C. and L.T. conducted the experiments. All authors analysed the data. L.M., L.Ø. and I.R. wrote the manuscript and all authors commented and edited the manuscript.

## Competing interests

The authors declare no competing interests.
