## [Peer Review File · Nature Communications]

REVIEWER COMMENTS

Reviewer #1 (Remarks to the Author):

The manuscript by Moubayidin et al. reports on SPT and HEC transcription factors regulating HD-ZIPII proteins, HAT3 and ATHB4 and that they are important for the bilateral to radial transition during gynoecium development. This paper is a continuation of previous work of the two groups. The Ostergaard's group showed the importance of SPT and HEC in regulating auxin distribution to be key for the radialization process in the gynoecium, the HD-ZIPII are added to be downstream of SPT/HEC. The Ruberti group showed a link with auxin and the HD-ZIPII proteins.

I have seen this manuscript around two years ago and all my comments made at that point (a long list) are basically all addressed. As mentioned before, the Scutt group showed in *Plant Cell* 2012 that the HD-ZIPII proteins are direct targets of SPT. The split-style phenotype in the *hat3 athb4* double mutant have already been described. There the authors already suggested a link with auxin. Furthermore, also the Lohmann group related HD-ZIPII proteins to HEC regulation. The authors improved a lot the genetics such as complementation experiments, crosses with marker lines with both mutants, etc. Especially part of the new results shown in Fig. 3 is a great addition making the whole story more complete. The text has been improved as well, including improved citations of the literature.

Although phenotypes and suggestions have been published before, the authors did a good job in the integration of all these pieces, together with a serious amount of new work, into a solid story proposing a model of bilateral to radial symmetry transition of the style of the gynoecium.

I like the actual story much more and it would be a good addition to the current literature. The manuscript is well written, and the data is of high quality and well presented in the figures. By reading this new manuscript I sadly noticed the recent loss of the last author, Ida Ruberti. Of course, academic quality goes first, and this manuscript clearly has the scientific quality, it would be a great thing to get this paper published in *Nature Communications*.

I do not have additional comments for the authors.

Best,
Stefan de Folter.

Reviewer #2 (Remarks to the Author):

Symmetry establishment and transition are very important processes in the development of multicellular organism and understanding the mechanisms underlying such processes is a very exciting goal for biologists. Authors address this question using the female reproductive structure of *Arabidopsis*. The *Arabidopsis* gynoecium initially emerges as a bilaterally symmetric structure reflecting the congenital fusion of two carpels. During development, the apical end undergoes a symmetry transition (from bilateral to radial) which passes through a transitory biradial stage. These transitions are guided by dynamic auxin distribution. In this paper, the authors show that SPATULA (SPT) and the three HECATE (HEC) proteins synergistically control the expression of the adaxial determinants HOMEODOMAIN ARABIDOPSIS THALIANA 3 (HAT3) and ARABIDOPSIS THALIANA HOMEODOMAIN 4 (ATHB4). These two HD-ZIP II genes act downstream of SPT and HEC and contribute to controlling the hormonal balance between auxin and cytokinin during axis formation and symmetry establishment in the gynoecium.

Concerning the novelty, the implication of SPATULA and HECATE genes in gynoecium morphogenesis and symmetry transition has already been studied in detail by Schuster and coll. (2015) who showed in particular the role of these genes in auxin distribution and sensitivity to cytokinin during gynoecium formation. The fact that auxin distribution exhibits dynamical pattern during symmetry transition in gynoecium has already been shown, including by the authors of the present paper in previous publications. HAT3 and ATHB4 have been identified as direct targets of SPATULA (Reymond and coll. (2012)). In addition, the implication of these two genes in establishing symmetry in body plan and auxin transport have already been shown (Turchi et al, 2013). In the end, the real new information presented in this paper is that the effect of SPT and HEC genes on gynoecium symmetry and development relies on the regulation of two HD-ZIP II genes, HAT3 and ATHB4. The precise mechanism by which HAT3 and ATHB4 control auxin

transport and cytokinin sensitivity has not been addressed in the present paper. For these reasons, I consider overstated to claim at the end of the abstract: " This work presents the first example of a biradial-to-radial symmetry switch in nature revealing the underlying molecular mechanism coordinating axes formation through genetic and hormonal interactions".

Concerning the work presented, authors have used very classical approaches. The results suffer from a few weaknesses among which the fact that no data are shown with the quadruple mutant *spt hec1,2,3* whereas this mutant is perfectly viable (see Schuster et al, 2015). In addition, some of the results are not supported by statistical analysis: the GUS staining shown in Figure 1 and Figure 3 are not supported by the number of samples analyzed and the dispersion (or absence of dispersion) of the results.

In conclusion, the work presented allows to link coherent pieces of information together, adding a new layer of regulators to a well-studied phenomena. This is of interest for biologists in the field but I think this work would be more adapted to a more specialized journal.

Reviewer #3 (Remarks to the Author):

The ms by Moubayadin et al expands on a previous story that showed how auxin dynamics in the growing gynoecium primordia was connected to correct morphogenesis and the transition from bilateral symmetry of the basal ovary and the radial symmetry of the apical style. This transition was dependent on the formation of four foci of auxin signaling (two medial and two lateral), that later were connected in a ring-shaped form by the depolarization of PIN transporters at the apex of the gynoecium driven by the concerted action of SPT and IND.

In this ms., new evidence is provided that led the authors to propose that the HEC factors are required together with SPT to regulate the activity of HAT3 and ATHB4 in the style, ultimately promoting the formation of the auxin ring at the apical end of the gynoecium and modulating CK sensitivity. While I believe that the data shown here have intrinsic value and are interesting, I am not convinced at all that they support the claims of the authors, since I think that they can fit into alternative models to the one proposed here (see below, but mainly, that the defects in adaxial-abaxial polarity in the *hat3 athb4* mutants impose physical constraints to the formation of a ring of auxin). I am also concerned about some of the writing, which could be sometimes a bit misleading. Some examples of this:

L38. The "This work presents the first example of biradial-to-radial symmetry switch". Not very sure about this sentence, since this was already stated in the previous paper and gives some false impression of novelty

L66. "It is unresolved how the radially symmetric ring-formed auxin maxima forms from the 4 foci". In the previous paper it is described that this happens by the loss of polarity of PIN transporters, which is caused by SPT-IND repression of PID. The question posed in the discussion of that paper is who is activating SPT-IND, nothing about ring formation, so I do not see how this sentence is truly valid.

Regarding the interpretation of results:

1. In my opinion, the phenotype of the double mutant *hat3 athb4* could be also interpreted by the physical constraints imposed by the defects in adaxial-abaxial polarity of the carpels. From the pics in Extended data fig2, it looks as if *hat3 athb4* mutants already have deformed gynoecium primordia as early as stage 5, and then, by stage 8-9 (Figure 2), that the four foci are physically separated by clefts that would make very difficult to form the auxin ring. Moreover, as the authors note (L72), the phenotype is similar to that of other mutants in adaxial-abaxial polarity (*jag nub*, for example), so it would appear that the role of HAT3 and ATHB4 in establishing radial symmetry is not specific. Moreover, in the apical gynoecium sections of *hat3 athb4* in Figure 1, it would appear that the medial protrusions of *hat3 athb4* have a pretty good radial symmetry themselves and a strong accumulation of auxin (Fig 2), arguing against their essential role in establishing radial form.

2. The authors show that driving HAT3 expression by the AS2 promoter in the *hat3 athb4*

background is sufficient to rescue the style clefts. Is AS2 expressed in the style like HAT3 and ATHB4? A detailed description of AS2 expression throughout gynoecium morphogenesis, or at least in similar stages to those shown in figure 1 would be necessary. If not, this would support the alternative explanation of general adaxial-abaxial defects in growth. If AS2 is expressed in the style, it would be important to determine whether the expression of HAT3 and ATHB4 in the apical domain of stage 8-9 gynoecia is essential by using a promoter specifically expressed there, and not in the adaxial domain of valves (SHI, for instance).

3. The authors treat with NPA the mutants to show complementation of style symmetry defects. This is suggestive, but not proof of specificity. Actually, in Staldal et al (2008) *New Phytol*, which has not been cited, it is shown that NPA treatments rescue split styles in mutants with very similar phenotypes to *hat3 athb4*, like *jag nub*, *lug* or *seu*, again arguing against a specific role of HAT3 and ATHB4 in the symmetry transition. It is likely that NPA cause pooling of auxin in the apical domain, creating a similar effect to the auxin ring.

4. The regulatory relationship of SPT and HEC with ATHB4 and HAT3 is clear. The strong decrease of HAT3 and ATHB4 expression in the apical domain of *hec spt* gynoecia and previously published ChIP data support their position downstream SPT/HEC. However, this does not necessarily mean that expression of HAT3 and ATHB4 in this domain is required for auxin ring formation (see previous comments). It would be crucial to check the PIN polarization state at relevant stages in the mutants, to check whether it is also depolarized (which was previously described as essential for ring formation), supporting the hypothesis, or not.

5. It is argued that the similarity of the *hec* style clefts with those of *athb4 hat3* styles support HAT3/ATHB4 as downstream effectors of HEC. However, this similarity is not so strong. *hec* triple mutants show very minor clefts, not always in the same positions (check Schuster et al or Gremski et al) and a pretty good radial style. Overinterpretation of these similarities should be avoided.

6. How is auxin distribution in lines where HAT3 or ATHB4 is induced? If overall levels increase significantly, pooling of auxin could mimic ring formation.

7. Finally, the model in Fig 4 places HAT3/ATHB4 downstream of SPT/IND. How does IND ox modify the *hat3 athb4* phenotype? If it does not rescue radial symmetry, it would also help to support the model.

Reviewer #1 (Remarks to the Author):

The manuscript by Moubayidin et al. reports on SPT and HEC transcription factors regulating HD-ZIP II proteins, HAT3 and ATHB4 and that they are important for the bilateral to radial transition during gynoecium development. This paper is a continuation of previous work of the two groups. The Ostergaard's group showed the importance of SPT and HEC in regulating auxin distribution to be key for the radialization process in the gynoecium, the HD-ZIP II are added to be downstream of SPT/HEC. The Ruberti group showed a link with auxin and the HD-ZIP II proteins.

I have seen this manuscript around two years ago and all my comments made at that point (a long list) are basically all addressed. As mentioned before, the Scutt group showed in Plant Cell 2012 that the HD-ZIP II proteins are direct targets of SPT. The split-style phenotype in the *hat3 athb4* double mutant have already been described. There the authors already suggested a link with auxin. Furthermore, also the Lohmann group related HD-ZIP II proteins to HEC regulation. The authors improved a lot the genetics such as complementation experiments, crosses with marker lines with both mutants, etc. Especially part of the new results shown in Fig. 3 is a great addition making the whole story more complete. The text has been improved as well, including improved citations of the literature.

Although phenotypes and suggestions have been published before, the authors did a good job in the integration of all these pieces, together with a serious amount of new work, into a solid story proposing a model of bilateral to radial symmetry transition of the style of the gynoecium.

I like the actual story much more and it would be a good addition to the current literature. The manuscript is well written, and the data is of high quality and well presented in the figures. By reading this new manuscript I sadly noticed the recent loss of the last author, Ida Ruberti. Of course, academic quality goes first, and this manuscript clearly has the scientific quality, it would be a great thing to get this paper published in Nature Communications.

I do not have additional comments for the authors.

Response: We are grateful to the reviewer for these supportive comments.

Reviewer #2 (Remarks to the Author):

Symmetry establishment and transition are very important processes in the development of multicellular organism and understanding the mechanisms underlying such processes is a very exciting goal for biologists. Authors address this question using the female reproductive structure of Arabidopsis. The Arabidopsis gynoecium initially emerges as a bilaterally symmetric structure reflecting the congenital fusion of two carpels. During development, the apical end undergoes a symmetry transition (from bilateral to radial) which passes through a transitory biradial stage. These transitions are guided by dynamic auxin distribution. In this paper, the authors show that SPATULA (SPT) and the three HECATE (HEC) proteins synergistically control the expression of the adaxial determinants HOMEODOMAIN ARABIDOPSIS THALIANA 3 (HAT3) and ARABIDOPSIS THALIANA HOMEODOMAIN 4 (ATHB4). These two HD-ZIP II genes act downstream of SPT and HEC and contribute to controlling the hormonal balance between auxin and cytokinin during axis formation and symmetry establishment in the gynoecium.

Concerning the novelty, the implication of SPATULA and HECATE genes in gynoecium morphogenesis and symmetry transition has already been studied in detail by Schuster and coll. (2015) who showed in particular the role of these genes in auxin distribution and sensitivity to cytokinin during gynoecium formation. The fact that auxin distribution exhibits dynamical pattern during symmetry transition in gynoecium has already been shown, including by the authors of the present paper in previous publications. HAT3 and ATHB4 have been identified as direct targets of SPATULA (Reymond and coll. (2012)). In addition, the implication of these two genes in establishing symmetry in body

plan and auxin transport have already been shown (Turchi et al, 2013). In the end, the real new information presented in this paper is that the effect of SPT and HEC genes on gynoecium symmetry and development relies on the regulation of two HD-ZIP II genes, HAT3 and ATHB4.

Response: We acknowledge that the novelty of the manuscript was not made sufficiently clear in the previous version of the manuscript. We have made major structural revisions to better emphasise the novelty, namely 1) the dynamics of auxin distribution (driven by process-specific TF combinations) underlies a step-wise establishment of polarity axes during gynoecium development and style radialisation, 2) that the HAT3/ATHB4 HD-ZIP II genes are induced by the second TF combination to coordinate adaxial-abaxial and medio-lateral polarity axes in a negative feedback loop with SPT, and 3) the activity of HAT3/ATHB4 coordinate the apical-basal and medio-lateral axes at least partially by controlling the hormonal balance between auxin and cytokinin.

To strengthen these points further, we present additional data that support a model by which HAT3 and ATHB4 exert dual control on auxin dynamics and axis formation. Firstly, our data suggest that the HDZIP-II control the upward auxin flux mediated by PIN1 to promote the adaxial-abaxial axis during style radialisation and sustaining the apical-basal axis in the ovary. Secondly, HAT3/ATHB4 repress SPT expression suggesting that they are involved in fine-tuning medio-lateral axis formation in a negative feedback loop. These new data have been added in the revised version of Figure 3, panels d and e.

The precise mechanism by which HAT3 and ATHB4 control auxin transport and cytokinin sensitivity has not been addressed in the present paper. For these reasons, I consider overstated to claim at the end of the abstract: " This work presents the first example of a biradial-to-radial symmetry switch in nature revealing the underlying molecular mechanism coordinating axes formation through genetic and hormonal interactions".

Response: We agree that this statement was imprecise. In the revision of the manuscript, we changed the abstract significantly and this statement is no longer included. Moreover, the new data included in Figure 3 address how HAT3 and ATHB4 control auxin transport, showing a specific effect on PIN1:GFP signal in the adaxial side of the mutant valves (Fig. 3d), correlating with low DR5rev::GFP signal in this tissue (Fig. 3a) and hypersensitivity to NPA applications (Fig. 3b,c and Suppl. Fig 7). In addition, our new data show HAT3 and ATHB4 feedback on SPT expression (Fig. 3e) possibly to finetune the appearance of the medial auxin foci (Fig. 3a).

We did attempt to address how the HD-ZIPs II might modulate cytokinin (CK) signalling by looking at the expression levels of specific (positive and negative) regulators of the CK signalling cascade, involved in gynoecium development and regulated by SPT (Reyes-Olalde et al. (2017) Plos Genet 13, e1006726). Our qRT-PCR showed that all A-type and B-type ARR_s tested were strongly upregulated in the hat3 athb4 background, meaning CK signalling is affected in this double mutant. Unfortunately, these data did not resolve how CK signalling is controlled HAT3 and ATHB4 at the tissue specific level; hence, we decided not to include these data in the revised version of the manuscript.

Concerning the work presented, authors have used very classical approaches. The results suffer from a few weaknesses among which the fact that no data are shown with the quadruple mutant *spt hec1,2,3* whereas this mutant is perfectly viable (see Schuster et al, 2015). In addition, some of the results are not supported by statistical analysis: the GUS staining shown in Figure 1 and Figure 3 are not supported by the number of samples analyzed and the dispersion (or absence of dispersion) of the results.

*Response: The synergistic activity of SPT and HECs in promoting HAT3 and ATHB4 expression was evident in vivo already from the analysis of pHAT3:GUS and pATHB4:GUS in various combinations of double and triple mutant between *spt* and *hec1,2,3* genotypes (Fig. 2b). Since the downregulation in*

transcription of the two HD-ZIPsII (i.e. molecular phenotype) was unveiled already in lower-order mutants, we propose that is unnecessary to include data on the spt,hec1,2,3 quadruple mutant here. Also, we have added in the text the number of samples analysed referring to the GUS expression analysis presented and provided statistical analysis for the hormonal treatments.

Reviewer #3 (Remarks to the Author):

The ms by Moubayadin et al expands on a previous story that showed how auxin dynamics in the growing gynoecium primordia was connected to correct morphogenesis and the transition from bilateral symmetry of the basal ovary and the radial symmetry of the apical style. This transition was dependent on the formation of four foci of auxin signaling (two medial and two lateral), that later were connected in a ring-shaped form by the depolarization of PIN transporters at the apex of the gynoecium driven by the concerted action of SPT and IND.

In this ms., new evidence is provided that led the authors to propose that the HEC factors are required together with SPT to regulate the activity of HAT3 and ATHB4 in the style, ultimately promoting the formation of the auxin ring at the apical end of the gynoecium and modulating CK sensitivity. While I believe that the data shown here have intrinsic value and are interesting, I am not convinced at all that they support the claims of the authors, since I think that they can fit into alternative models to the one proposed here (see below, but mainly, that the defects in adaxial-abaxial polarity in the hat3 athb4 mutants impose physical constrains to the formation of a ring of auxin). I am also concerned about some of the writing, which could be sometimes a bit misleading. Some examples of this:

L38. The “This work presents the first example of biradial-to-radial symmetry switch”. Not very sure about this sentence, since this was already stated in the previous paper and gives some false impression of novelty

Response: In the paper by Moubayidin and Østergaard (2014 Current Biology) which the reviewer refers to, we described the role of auxin dynamics to the overall bilateral-to-radial process (not biradial-to-radial). Nevertheless, we have removed this sentence in the revised manuscript.

L66. “It is unresolved how the radially symmetric ring-formed auxin maxima forms from the 4 foci”. In the previous paper it is described that this happens by the loss of polarity of PIN transporters, which is caused by SPT-IND repression of PID. The question posed in the discussion of that paper is who is activating SPT-IND, nothing about ring formation, so I do not see how this sentence is truly valid.

Response: We agree that this sentence could be misinterpreted and it is no longer present in the revised manuscript. Instead, we have strived to strengthen and clarify that the aim of this manuscript is to unveil how the dynamics of auxin distribution (driven by process-specific TF combinations) underlies a step-wise establishment of the axes during style radialisation. We believe, our new findings are coherently integrated onto the carefully orchestrated series of events driven by auxin important for the apical-basal (Nemhauser et al 2000) and medio-lateral (Larsson et al 2014; Moubayidin and Østergaard, 2014) axis establishment.

Regarding the interpretation of results:

1. In my opinion, the phenotype of the double mutant hat3 athb4 could be also interpreted by the physical constrains imposed by the defects in adaxial-abaxial polarity of the carpels. From the pics in Extended data fig2, it looks as if hat3 athb4 mutants already have deformed gynoecium primordia as

early as stage 5, and then, by stage 8-9 (Figure 2), that the four foci are physically separated by clefts that would make very difficult to form the auxin ring. Moreover, as the authors note (L72), the phenotype is similar to that of other mutants in adaxial-abaxial polarity (jag nub, for example), so it would appear that the role of HAT3 and ATHB4 in establishing radial symmetry is not specific. Moreover, in the apical gynoecium sections of *hat3 atb4* in Figure 1, it would appear that the medial protrusions of *hat3 atb4* have a pretty good radial symmetry themselves and a strong accumulation of auxin (Fig 2), arguing against their essential role in establishing radial form.

Response: The reviewer raises several interesting issues here regarding the roles of HAT3/ATHB4, but we believe some may have arisen due to misunderstandings and we have paid special attention to clarify these points in the revised manuscript.

*Firstly, HAT3/ATHB4 promote adaxial identity (e.g. Turchi et al. (2013) Development 140, 2118-29), and as such they promote one step of the overall symmetry transition taking place at the gynoecium apex, which requires coordination of the adaxial-abaxial axis and is established after the apical-basal (Nemhauser et al 2000) and medio-lateral (Larsson et al 2014) axes. To support this role, we complemented the *hat3 atb4* mutant phenotype by expressing HAT3 specifically under control of the adaxial promoter AS2 (Fig. 1d). Moreover, it is also true that other polarity regulators function in symmetry formation of the apical gynoecium. However, we do not consider this an issue in terms of revealing the role of HAT3/ATHB4 and it is outside the scope of this manuscript to establish how and if these factors function in the same pathway.*

*Secondly, we thank the reviewer for noticing that “the medial protrusions of *hat3 atb4* have a pretty good radial symmetry themselves”. In this revised version we provide evidence that highlight a previously unrecognized negative feedback loop mechanism operated by the HD-ZIPs-II fundamental for the coordination between the medial-lateral and the adaxial-abaxial axes: We find that the adaxial factors HAT3 and ATHB4 repress the expression of the medial factor SPT (Fig. 3e), which ultimately is responsible for: 1) the formation of radially symmetric growth in the medial domain of *hat3 atb4*, as shown by the triple *hat3 atb4 spt* mutant which displays clefts positioned at the medial apical region, similarly to *spt* and opposite to *hat3 atb4* (Fig. 2c) mutants; and 2) the early auxin accumulation in the medial foci of *hat3 atb4* (Fig. 3a).*

2. The authors show that driving HAT3 expression by the AS2 promoter in the *hat3 atb4* background is sufficient to rescue the style clefts. Is AS2 expressed in the style like HAT3 and ATHB4? A detailed description of AS2 expression throughout gynoecium morphogenesis, or at least in similar stages to those shown in figure 1 would be necessary. If not, this would support the alternative explanation of general adaxial-abaxial defects in growth. If AS2 is expressed in the style, it would be important to determine whether the expression of HAT3 and ATHB4 in the apical domain of stage 8-9 gynoecia is essential by using a promoter specifically expressed there, and not in the adaxial domain of valves (SHI, for instance).

Response: We appreciate the suggestions by the reviewer. However, we disagree that these experiments are necessary for the conclusions made in this work. Instead, we have modified the text to say: “These data show that HAT3 and ATHB4 HD-ZIP II proteins are required to coordinate growth along the medio-lateral polarity axis during radialisation of the style, functioning at least partially in a cell-autonomous manner from adaxial domains.”

3. The authors treat with NPA the mutants to show complementation of style symmetry defects. This is suggestive, but not proof of specificity. Actually, in Staldal et al (2008) New Phytol, which has not been cited, it is shown that NPA treatments rescue split styles in mutants with very similar phenotypes to *hat3 atb4*, like *jag nub*, *lug* or *seu*, again arguing against a specific role of HAT3 and ATHB4 in the symmetry transition. It is likely that NPA cause pooling of auxin in the apical domain, creating a similar effect to the auxin ring.

Response: We agree that a reference to Ståldal et al ((2008) New Phytologist 180, 798-808) was missing and it is now included. As above, we are confused with the reviewer's focus on specificity. We are not claiming that HAT3/ATHB4 are uniquely responsible for the biradial-to-radial conversion. We show that the SPT/HEC module promotes HAT3/ATHB4 expression and that this regulatory process is required for the adaxial-abaxial axis to be integrated at the last stage of style symmetry establishment, following on from the apical-basal and the medial-lateral axes, in a precise spatial and temporal manner.

Moreover, in Ståldal et al (2008) the authors concluded the following: "This suggests that elevated apical auxin concentrations [following NPA application] can compensate for the loss of a large number of style-promoting factors and that auxin may act downstream of, or in parallel with, these." This conclusion is in line with our conclusion that HAT3 and ATHB4 downstream control auxin polar transport. Moreover, the rescue of the hat3 athb4 phenotype observed following NPA application of inflorescences shows that nearly 100% of styles become radial, compared to 62% of radial styles observed in spt-12 mutants treated with NPA (Fig. 3c). This hypersensitivity of hat3 athb4 in the apical region as well as along the apical-basal axis (Suppl. Fig. 7) is similar to that observed after NPA treatments of lug-1 and seu-1 mutant gynoecia and specifically different from the phenotype observed in NPA-treated nub1 jag-1 (Ståldal et al. 2008), thus arguing in favour of a role for HAT3 and ATHB4 as axes coordinators via controlling auxin dynamics.

4. The regulatory relationship of SPT and HEC with ATHB4 and HAT3 is clear. The strong decrease of HAT3 and ATHB4 expression in the apical domain of *hec spt* gynoecia and previously published CHIP data support their position downstream SPT/HEC. However, this does not necessarily mean that expression of HAT3 and ATHB4 in this domain is required for auxin ring formation (see previous comments). It would be crucial to check the PIN polarization state at relevant stages in the mutants, to check whether it is also depolarized (which was previously described as essential for ring formation), supporting the hypothesis, or not.

Response: We are grateful for these comments which led us to carry out additional experiments and refining our model. The data are presented in the revised Figure 3 and support a model by which HAT3 and ATHB4 exert dual control on auxin dynamics and axis formation. Firstly, our data suggest that the HDZIP-IIIs control the upward auxin flux mediated by PIN1 to promote the adaxial-abaxial axis during style radialisation and sustaining the apical-basal axis in the ovary. Secondly, HAT3/ATHB4 repress SPT expression suggesting that they are involved in fine-tuning medio-lateral axis formation in a negative feedback loop. The new data included in Figure 2 show that HAT3 ATHB4 control PIN1::GFP in the adaxial side of the valves (Fig. 3d), correlating with low DR5rev::GFP signal in the valves (Fig. 3a) and hypersensitivity to NPA application (Fig. 3c and Suppl. Fig. 7). We did not observe significant variation in the cellular apolar distribution of PIN1::GFP in the apical region of the hat3 atbh4 mutant (data not shown). This is not unexpected, given the ectopic expression of SPT in this background (Fig. 3e).

5. It is argued that the similarity of the *hec* style clefts with those of *athb4 hat3* styles support HAT3/ATHB4 as downstream effectors of HEC. However, this similarity is not so strong. *hec* triple mutants show very minor clefts, not always in the same positions (check Schuster et al or Gremski et al) and a pretty good radial style. Overinterpretation of these similarities should be avoided.

Response: We have made major modifications to the manuscript including restructuring and rewriting passages. We have taken this comment into account.

6. How is auxin distribution in lines where HAT3 or ATHB4 is induced? If overall levels increase significantly, pooling of auxin could mimic ring formation.

Response: As far as we are aware, there is no evidence to suggest that HAT3/ATHB4 induce auxin biosynthesis. We are not set up to carry out auxin measurements in our labs and would have to do this by collaboration. Collecting enough induced tissue and send it on dry ice to a collaborator would

be tricky under the current working conditions and we therefore hope the referee will understand why this experiment has not been done.

As additional evidence that HAT3/ATHB4 regulate auxin distribution, we hope the PIN1:GFP in hat3 athb4 background (see above) and the premature emergence of the medial auxin foci at early developmental stages of the hat3 athb4 mutant, now included in this revised version of the manuscript, will help to strengthen this point.

7. Finally, the model in Fig 4 places HAT3/ATHB4 downstream of SPT/IND. How does IND ox modify the hat3 athb4 phenotype? If it does not rescue radial symmetry, it would also help to support the model.

Response: It is true that the model in Fig. 4 would predict that overexpression of IND is unable to rescue the defects in the hat3 athb4 double mutant. However, this experiment would be difficult to perform as both hat3 athb4 loss-of-function mutations and ectopic expression of IND have devastating effects on plant development. As we have previously published, simple 35S::IND overexpressor lines do often not reach the reproductive stage, but instead terminate by producing pin-like inflorescences (Sorefan et al. (2009) Nature 459, 583-86). Although, we do have an inducible version of IND (35S::IND-GR), it would not be feasible to apply the inducing chemical (Dex) strictly to the developing hat3 athb4 mutant inflorescences, owing to the pleiotropic defects in organ patterning and growth. Producing an inducible version driven under a tissue-specific promoter (e.g. AS2) would be possible; however, we know from experience that any leaky expression of IND has dramatic effects that would likely make it difficult to interpret the results.

Furthermore, since SPT is upregulated by overexpression of IND (Girin et al., (2011) Plant Cell 23, 3641-53) and both transcription factors work synergistically in symmetry establishment, the new findings presented in this revised version, i.e. HAT3/ATHB4 repress SPT expression, suggests the effect of the IND-mediated upregulation of SPT would enhance the hat3athb4 mutant defects at the medial axis, which we already proved to be dependent on SPT activity in the hat3 athb4 spt triple mutant background (Fig. 2c).

Therefore, under the COVID pandemic, we have had to carefully prioritise experiments and preparation of lines. Given that such an experiment – if it worked – would not provide significant new information, we hope that the reviewer can accept our suggestion not to carry it out.

REVIEWERS' COMMENTS

Reviewer #2 (Remarks to the Author):

I acknowledge the efforts made by the authors to address the comments of the review and to modify the manuscript accordingly. The results reported are convincing and taken together with already published data, they make a nice and coherent story of interest for people in the field. Still, I am not convinced that the main message is of sufficient general interest to be published in Nature Communications.

Reviewer #3 (Remarks to the Author):

The ms by Carabelli et al is a resubmission, quite modified in content, of a previous work that I also revised. The new version takes into account several of the comments that another reviewer and myself did, revisiting the data and toning down several of the conclusions, which now are more focused not so much on the role of ATHB4 and HAT3 in the transition from the biradial-to-radial symmetry in the style, but on the coordination of adaxial-abaxial polarity with the radialization of the style, downstream of SPT/HEC.

I believe that the data in the manuscript now are more consistent with the results, but that again, they fall a bit short to draw a really novel story. One thing that I find critical is that the lack of proper style formation in the *athb4 hat3* mutants can be explained by the defects in the adaxial-abaxial polarity (which cause clefts and maybe failure to form a ring just because of physical constraints), and not so much by a specific role downstream of SPT/HEC in the style. The results shown here demonstrate a clear reduction of ATHB4/HAT3 expression in the style of *spt/hec* mutants, but just driving HAT3/ATHB4 with the AS2 promoter rescues all defects. If AS2 is not expressed in the style, this argues against the model proposed by the authors, supporting instead that restoring adaxial-abaxial polarity in early stages of gynoecium development would be enough (required, as the authors say, L119, but ALSO sufficient) independently of SPT/HEC. I asked for these experiments (really easy to do) but they have not been provided. It is also not shown whether the adaxial expression of HAT3 and ATHB4 is changed at early stages in the adaxial domain of *spt/hec* mutants, an important result to discuss the model properly, and to reveal the relative importance of the late regulation of ATHB4/HAT3 by SPT/HEC vs early effects.

New data are provided on auxin dynamics in the *athb4 hat3*, mainly through PIN regulation, but this is not really new, just more precise in the context of the gynoecium. Also, a negative feedback loop on SPT is revealed, which is interesting, but somehow incomplete, since overexpression of SPT is known to cause no phenotypic effects in gynoecium development, so it is difficult to see the importance of this fine-tune regulation without bringing more players yet to be uncovered.

Please find below our point-by-point response to the reviewer comments:

Reviewer #2 (Remarks to the Author):

I acknowledge the efforts made by the authors to address the comments of the review and to modify the manuscript accordingly. The results reported are convincing and taken together with already published data, they make a nice and coherent story of interest for people in the field. Still, I am not convinced that the main message is of sufficient general interest to be published in Nature Communications.

We are glad Reviewer 2 appreciated the revised version of our manuscript, which is accompanied by new supportive data. We believe our study and model system have revealed a hitherto unappreciated stepwise recruitment of components that control body-axes formation during organ symmetry transition. Hence, our findings will be of general importance for scientists interested in developmental genetics.

Reviewer #3 (Remarks to the Author):

The ms by Carabelli et al is a resubmission, quite modified in content, of a previous work that I also revised. The new version takes into account several of the comments that another reviewer and myself did, revisiting the data and toning down several of the conclusions, which now are more focused not so much on the role of *ATHB4* and *HAT3* in the transition from the biradial-to-radial symmetry in the style, but on the coordination of adaxial-abaxial polarity with the radialization of the style, downstream of SPT/HEC.

We are glad Reviewer 3 appreciated the focus of our manuscript has been devoted to highlight the biological relevance of the molecular framework presiding over style radialization.

I believe that the data in the manuscript now are more consistent with the results, but that again, they fall a bit short to draw a really novel story. One thing that I find critical is that the lack of proper style formation in the *athb4 hat3* mutants can be explained by the defects in the adaxial-abaxial polarity (which cause clefts and maybe failure to form a ring just because of physical constraints), and not so much by a specific role downstream of SPT/HEC in the style.

We showed that overexpression of *HAT3* or *ATHB4* can rescue *spt* style defects (Fig. 2d,e), which are not related to the adaxial-abaxial polarity, meaning these HD-ZIPs-II can work downstream of SPT in radial symmetry establishment (please, also see the next response). Moreover, the additive phenotype of the *spt hat3 athb4* triple mutant rules out the hypothesis that physical constraints and conflicts in tissue growth between the medial and adaxial axes are causative of the split style phenotype observed in *hat3 athb4*, since removing SPT activity in the *hat3 athb4* background did not rescue the symmetry brake in the double mutant styles. If defects in the adaxial tissues caused physical constraints during gynoecium development, one would expect developmental defects to vastly extend throughout the ovary, while the clefts develop specifically at the uppermost end of the *hat3 athb4* mutant gynoecia.

The results shown here demonstrate a clear reduction of *ATHB4/HAT3* expression in the style of *spt/hec* mutants, but just driving *HAT3/ATHB4* with the AS2 promoter rescues all defects. If AS2 is not expressed in the style, this argues against the model proposed by the authors, supporting instead that restoring adaxial-abaxial polarity in early stages of gynoecium

development would be enough (required, as the authors say, L119, but ALSO sufficient) independently of SPT/HEC. I asked for these experiments (really easy to do) but they have not been provided. It is also not shown whether the adaxial expression of HAT3 and ATHB4 is changed at early stages in the adaxial domain of *spt/hec* mutants, an important results to discuss the model properly, and to reveal the relative importance of the late regulation of ATHB4/HAT3 by SPT/HEC vs early effects.

In the Source data file we have provided an additional example of *pAS2:HAT3:YFP* expression, which shows signal in the adaxial side of the valve extending along the apical-basal axis, including the apical end where the style develops.

Also, it is important to notice that overexpression of HAT3 and ATHB4 does not cause patterning changes during gynoecium development (Suppl. Fig. 6), similarly to the overexpression of SPT. Although, overexpressing these HD-ZIPs-II in the *spt* mutant background can overcome the developmental stall displayed by the mutant, and produce radially symmetric styles (Fig. 2d,e). This data is coherent with our working model where SPT is both upstream and downstream of HAT3 and ATHB4, in a negative feedback loop mechanism.

In other words, the overexpression of either SPT or HAT3/ATHB4 is not sufficient to trigger ectopic radialization of the gynoecium, but it is precisely the correct timing of expression of the SPT/HECs module inducing *HD-ZIPs-II* expression, and the subsequent repression of *SPT* expression by HAT3 and ATHB4 that is sufficient to determine the transition.

New data are provided on auxin dynamics in the *athb4 hat3*, mainly through PIN regulation, but this is not really new, just more precise in the context of the gynoecium. Also, a negative feedback loop on SPT is revealed, which is interesting, but somehow incomplete, since overexpression of SPT is known to cause no phenotypic effects in gynoecium development, so it is difficult to see the importance of this fine-tune regulation without bringing more players yet to be uncovered

We acknowledge no defects have been previously reported following the overexpression of SPT in a wild-type background; This might be due to the requirement for either a protein partner or a post-translational control mechanism, which might be rate-limiting the activity of SPT. A different scenario might be occurring in the *hat3 atbh4* mutant background, where we provided evidence that high levels of *SPT* transcription (Fig. 3e) leads to sustained radial growth in the medial-apical region of the *hat3 atbh4* double mutant (Fig. 2c).